# Comprehensive Management of Blood Pressure in Patients with Septic AKI

**DOI:** 10.3390/jcm12031018

**Published:** 2023-01-28

**Authors:** Junhui Deng, Lina Li, Yuanjun Feng, Jurong Yang

**Affiliations:** 1Department of Nephrology, The Third Affiliated Hospital of Chongqing Medical University, Chongqing 400120, China; 2Department of Renal Rheumatology, Space Hospital Affiliated to Zunyi Medical University, Zunyi 563002, China

**Keywords:** acute kidney injury, sepsis, blood pressure, prognosis, microcirculation

## Abstract

Acute kidney injury (AKI) is one of the serious complications of sepsis in clinical practice, and is an important cause of prolonged hospitalization, death, increased medical costs, and a huge medical burden to society. The pathogenesis of AKI associated with sepsis is relatively complex and includes hemodynamic abnormalities due to inflammatory response, oxidative stress, and shock, which subsequently cause a decrease in renal perfusion pressure and eventually lead to ischemia and hypoxia in renal tissue. Active clinical correction of hypotension can effectively improve renal microcirculatory disorders and promote the recovery of renal function. Furthermore, it has been found that in patients with a previous history of hypertension, small changes in blood pressure may be even more deleterious for kidney function. Therefore, the management of blood pressure in patients with sepsis-related AKI will directly affect the short-term and long-term renal function prognosis. This review summarizes the pathophysiological mechanisms of microcirculatory disorders affecting renal function, fluid management, vasopressor, the clinical blood pressure target, and kidney replacement therapy to provide a reference for the clinical management of sepsis-related AKI, thereby promoting the recovery of renal function for the purpose of improving patient prognosis.

## 1. Introduction

Septic shock is a pathological syndrome of life-threatening multiorgan dysfunction caused by a dysregulated host response to infection [1]. Acute kidney injury (AKI) occurs in up to 50% of patients with sepsis, resulting in prolonged hospitalization and a six- to eightfold increase in mortality [2,3,4]. Survivors of AKI are at risk of developing chronic or end-stage renal disease, even with short-term or mild renal impairment, which is a medical burden on society [5,6]. Multiple mechanisms are involved in the development of sepsis-associated AKI, including inflammatory responses, oxidative stress, adaptive responses of renal tubular epithelial cells, and renal hemodynamic abnormalities [7,8,9]. Histological staining of renal tissue from patients who died of sepsis-associated AKI reveals uneven tubular injury, manifested as focal acute tubular injury, and most of the renal tubules were normal [10,11]. This is related to abnormal shunt of renal blood supply due to microcirculation damage, and clinical improvement of blood pressure and optimization of systemic hemodynamic support are essential in patients with AKI or at risk of AKI [12,13]. Statistically, a high proportion of patients diagnosed with sepsis-associated AKI in the intensive care unit have previous chronic hypertension, which is an independent risk factor [14,15]. Because the kidneys of hypertensive patients are more sensitive to abnormal fluctuations in blood pressure [16,17], the management of blood pressure in septic AKI is a great challenge for clinicians. Therefore, this paper summarizes the mechanism of sepsis microcirculation disorder affecting renal function, clinical comprehensive management of blood pressure, and target blood pressure in order to provide references for clinical treatment and urgent clinical research directions.

## 2. Microcirculatory Disorders

Microcirculation refers to the capillary system that connects the arteries and veins. It is essential to the function of every organ and system in the body for the transport of oxygen and nutrients and the removal of toxins. Impaired microcirculation is a hallmark of septic shock and is the pathophysiological basis for the failure of multiple organs, including the heart, intestines, liver, brain, and kidneys [18,19]. The occurrence of microcirculatory disorders in sepsis involves multiple mechanisms related to endothelial cell injury, blocked intercellular communication, glycocalyx shedding, coagulation dysfunction, leukocyte and platelet adhesion, and altered red blood cell deformability due to severe infection, resulting in reduced microvascular blood flow velocity and microthrombosis, which ultimately disrupt microvascular flow [20,21]. The kidney is particularly rich in microvessels, and impaired microcirculation leads to intrarenal perfusion redistribution, resulting in abnormal blood flow distribution in the renal medulla and renal cortex and decreased oxygenation in tissues, which triggers a vicious cycle of oxidative stress and inflammation [22,23,24]. There is a natural shunt in renal blood flow, with cortical blood perfusion accounting for 80% of renal blood flow (RBF) and a partial pressure of oxygen of approximately 70 mmHg and medullary blood flow accounting for 20% of RBF and a partial pressure of oxygen of approximately 20 mmHg. Compared with the cortex, the medullary circulation is less able to self-regulate against ischemia and hypoxia, making the medulla more sensitive to hypoxia caused by sepsis [25]. This is related to the specific topography of the vascular bundle in the renal medulla, where the thick upper branches of the Henle collaterals are located at the periphery of the vascular bundle, allowing oxygen diffusion to be restricted. In addition, diffuse shunting of oxygen in the renal medulla from the straight ducts down to the upper ducts may reduce oxygen transport [26,27]. Reduced blood flow and decreased oxygenation in the renal medulla may exacerbate tubular epithelial cell injury in the medullary portion, especially in the proximal tubule S3 segment in the outer medulla, leading to increased reactive oxygen species and mitochondrial dysfunction, further leading to reduced renal function [28,29,30].

Hypoxia in the medulla also exacerbates renal injury by involving the tubuloglomerular feedback mechanism (TGF). Hypoxia in the thick ascending branch of the Henle collaterals can lead to decreased production of adenosine triphosphate, which decreases sodium reabsorption, activates TGF, and reduces the glomerular filtration rate by constricting the afferent small arteries, which again triggers a vicious cycle of medullary hypoperfusion and hypoxia [31]. In conclusion, abnormal renal medullary perfusion due to microcirculatory impairment, tissue ischemia and hypoxia, and reaggravation of medullary hypoxia by TGF in sepsis are the main pathological mechanisms for the development of AKI associated with sepsis (Figure 1), and renal medullary hypoxia can be corrected clinically by improving microcirculation, thereby reversing renal impairment.

## 3. Fluid Management

In clinical practice, intractable hypotension due to sepsis is preferentially resuscitated by fluid therapy to achieve the blood pressure target, thereby restoring circulating volume and improving renal perfusion [32]. A retrospective study defined resuscitation goals for patients with sepsis in the ICU as mean arterial pressure greater than 65 mmHg, central venous pressure greater than 8 mmHg, and central venous oxygenation greater than 70%; achieving early resuscitation within 6 h after consultation was associated with a reduction in the development of AKI [33]. The main categories of volume expansion fluid therapy for hypotensive shock are colloids and crystals, and a meta-analysis found that balanced crystalloids and albumin reduced the mortality in patients with sepsis more than hydroxyethyl starch (HES) and saline [34]. The results of this study showed that HES is associated with a higher incidence of AKI and renal replacement (RRT) rate, leading to the recommendation that HES solution be contraindicated in patients with severe sepsis or at risk of AKI [35,36,37]. In a sheep model of septic shock, renal function and cumulative diuresis were preserved in the albumin and crystalloid resuscitation groups, whereas HES resulted in reduced creatinine clearance [38]. However, given the risk of infection, 4% hypotonic albumin is recommended as the concentration [39].

Additionally, 0.9% physiological saline was found to be at risk of causing or exacerbating metabolic acidosis and progression to AKI and RRT in critically ill patients with sepsis [40,41]. However, most patients with sepsis-related AKI experience fluid overload to varying degrees during fluid resuscitation therapy, which can aggravate renal impairment and increase the morbidity and mortality of patients [42,43,44]. In addition to blood pressure, CVP is one of the main clinical indicators of hemodynamics. It usually refers to the pressure in the right atrium and large intrathoracic veins and is important for understanding the effective circulating blood volume and cardiac function [45,46]. A cohort study that included 15 patients with sepsis at the main time indicated that elevated CVP was associated with an increased risk of death and AKI, and in addition, each 1 mmHg increase in CVP was associated with a 6% increase in the odds of AKI [47]. Huo et al. found that a lower CVP level (<13 mmHg) was an independent variable associated with reduced mortality in patients with sepsis-related AKI through a propensity-score-matched analysis of clinical data on adult sepsis from the Medical Information Mart for Intensive Care-IV database [48]. To address fluid overload, clinicians have three primary options: fluid restriction, diuretic pharmacotherapy, or RRT extracorporeal ultrafiltration. Collectively, diuretics should not be used to treat AKI except for fluid overload [49]. Diuretic clearance fluid is harmful in the acute phase of sepsis, and when the patient is stable, fluid overload is harmful and diuretic clearance can be applied [50,51,52]. Kidney replacement therapy (RRT) is an effective treatment for dehydration in patients with hemodynamic instability where fluid overload is not appropriate for diuretics. In conclusion, fluid resuscitation is the primary means to improve sepsis patients and patients with concurrent AKI. In clinical practice, it is necessary to avoid rehydration therapy that is detrimental to the benefit of the kidney, while strictly vigilant against fluid overload. When fluid overload occurs, diuretic medication or RRT may be selected, depending on the patient’s situation.

## 4. Vasopressor

In septic hypotension, if the blood pressure target cannot be achieved by fluid augmentation, clinicians can also implement resuscitation to improve the microcirculatory perfusion of the kidneys and other organs through the use of a vasopressor. Currently, the commonly used drugs include norepinephrine, dopamine, pressor, and angiotensin II (Table 1).

Norepinephrine increases the arterial pressure through α-adrenergic receptor-mediated vasoconstriction and is the medication of choice for septic hypotension [53,54,55]. Infusion of norepinephrine in a hyperdynamic sheep model of sepsis increased blood flow to the heart, intestine, and kidneys, effectively increasing urine output and improving creatinine clearance [56]. Although it has been reported that norepinephrine can aggravate medulla hypoxia while reviving blood pressure in a sheep sepsis AKI model, it has no significant effect on renal blood flow or renal oxygen delivery [57,58]. Furthermore, systematic reviews and metastudies have found that norepinephrine minimizes arrhythmia compared with other vasopressors and is therefore safe for use in patients with septic shock and AKI [53].

Dopamine is a natural precursor of epinephrine and norepinephrine. It excites mainly the α receptors, β receptors, and peripheral dopamine receptors, and it has shown comparable efficacy with epinephrine in the treatment of infectious shock [59]. Due to its stimulatory effect on dopaminergic receptors, dopamine is suspected to adversely affect the perfusion output of internal organs, such as the kidney [60]. In a prospective, double-blind randomized controlled study, “low-dose” dopamine worsened renal perfusion and increased renal impairment in patients with acute kidney failure in the absence of systemic hemodynamic effects [61]. Dopamine use in septic hypotensive patients was associated with higher rates of mortality, infection, and arrhythmic events compared with norepinephrine and epinephrine [62,63,64,65,66].

Vasopressin increases arterial blood pressure primarily by stimulating the arginine vasopressin receptor 1A (AVPR1A) located on vascular smooth muscle cells to induce vasoconstriction. It is currently recommended as a pressor agent that is unresponsive to norepinephrine, and/or it reduces the dose of norepinephrine required to achieve the blood pressure target [67,68].

In a sheep septic AKI model, vasopressin had better renoprotective effects than norepinephrine [69,70]. In clinical practice, vasopressin increases per beat output, improves renal perfusion in patients with infectious shock, improves renal function, and reduces the use of RRT [71,72]. However, it has also been found that in adults with infectious shock, early administration of vasopressin did not improve the number of days without kidney failure compared with norepinephrine [73]. Therefore, the benefit of vasopressin in septic AKI also needs to be informed by more studies.

Angiotensin II, an endogenous circulating hormone with potent vasoconstrictive effects, effectively increases blood pressure in patients with vasodilatory shock who do not respond to high doses of conventional vasopressors, such as norepinephrine [74]. The decrease in renal arteriolar pressure and sodium in protourine stimulates the secretion of renin by periglomerular cells. Renin acts on angiotensinogen in plasma to produce inactive angiotensin I, which is hydrolyzed to active angiotensin II under the action of angiotensin-converting enzyme. Angiotensin Ⅱ can cause the contraction of the afferent and efferent arterioles, and the efferent arterioles are more sensitive, which increases the pressure in the glomeruli, thus affecting the glomerular filtration rate [75,76]. In a sheep model of septic AKI, angiotensin II was found to restore arterial pressure without exacerbating medullary hypoxia. It significantly increased urine output and normalized creatinine clearance [77,78]. Clinically, in patients with septic acute kidney injury requiring kidney replacement therapy, the 28-day survival and mean arterial pressure response were higher in the angiotensin II group compared with the placebo group, and the release rate from kidney replacement therapy was higher, suggesting that patients with septic AKI requiring kidney replacement therapy may preferentially benefit from angiotensin II [79]. Taken together, we can conclude that angiotensin II appears to be a safe and effective treatment, although more clinical data are still needed to support this.

**Table 1 jcm-12-01018-t001:** Pharmacological effects and renal effects of the vasopressors.

Drug Name	Receptor	Effects on Renal Function	Effect on Urine Output	Effects on the Medulla	Reference
Norepinephrine	α-adrenergic receptor	Improve	Increase	Exacerbate hypoxia	[53,54,55,56,57,58]
Dopamine	α receptor, β receptor, and peripheral dopamine receptor	Worsen	Unknown	Unknown	[60,61,80,81]
Vasopressin	AVPR1A	Improve	Increase	Unknown	[67,68,69,70,71,72]
Angiotensin II	Angiotensin II receptor	Improve	Increase	Does not exacerbate hypoxia	[74,77,78]

AVPR1A, arginine vasopressin receptor 1A.

## 5. Blood Pressure Target

Sepsis often leads to intractable hypotension, resulting in hypoperfusion and microcirculatory disorders in multiple organs, including the kidneys. Aggressive clinical correction of blood pressure can largely improve renal blood supply and restore renal function [32,82,83]. Mean arterial pressure (MAP) is the average arterial blood pressure during one cardiac cycle. It is widely used as a measure of blood pressure. The latest guidelines on sepsis suggest a MAP garget of 65 mmHg for initial resuscitation in patients with septic shock to reduce the risk of death and end-organ failure [84]. However, it is important to note that even at the same MAP, organs may have different perfusion pressures and pressure–flow rates, with the small postglomerular artery responsible for renal tissue perfusion typically having a lower pressure than systemic arterial blood pressure [85]. Therefore, for patients with sepsis, there are different recommendations for the ideal blood pressure target to help prevent renal impairment or aid recovery (Table 2). A retrospective study from 110 hospitals in the United States evaluated patients with sepsis who were admitted to the intensive care unit (ICU) for more than 24 h from 2010 to 2016. Through multivariate logistic regression analysis, the study found that the time-weighted mean arterial pressure (TWA-MAP) began to be significant for AKI at 65 mmHg, and as MAP decreased from 85 to 55 mmHg, in addition to AKI, patients had a progressive increase in mortality and odds of myocardial injury. Therefore, it is suggested that in septic patients, maintaining MAP well above 65 mmHg may be a more prudent approach to prevent and ameliorate renal injury [86]. In another prospective observational FINNAKI study from 17 hospitals in Finland, 423 patients with severe sepsis from September 2011 to February 2012 were enrolled by screening, of whom 153 (36.2%) presented with AKI, and a significantly higher TWA-MAP of 78.6 mmHg (72.9–85.4 mmHg) was found by between-group analysis in non-AKI patients than in AKI patients. The study indicated that a TWA-MAP cutoff of 73 mmHg best predicted the progression of sepsis-related AKI [87]. Chen et al. conducted a retrospective cohort study of patients admitted to the ICU of Chiayi Chang Gung Memorial Hospital from January 2015 to December 2016, which included 63 critically ill patients with confirmed AKI. They found that the sensitivity and specificity of the total patient mortality at MAP ≤ 77.16 mmHg were 62.50% and 91.30%, respectively, and they concluded that MAP ≤ 77 mmHg can be used as a risk factor for death in patients with AKI [88]. A prospective study, which was approved by the Association des Réanimateurs du Centre-Ouest, France, conducted in two ICUs from October 2007 to April 2009, included and followed 217 patients with septic shock and compared the MAP in the first 24 h after inclusion in patients who developed AKI at 72 h and those who did not develop AKI. The study found that the optimal MAP level to prevent AKI at 72 h after the onset of infectious shock was between 72 and 82 mmHg [89]. A retrospective single-center cohort study from the Mayo Clinic found the lower incidence of AKI in patients with infectious shock when the postresuscitation median MAP was closest to or higher than the preadmission median MAP, suggesting that this MAP should not be a fixed value and that it is necessary to target the preadmission MAP for AKI prevention [90].

For patients with septic shock with a history of hypertension, European experts suggest personalized blood pressure targets, perhaps with a higher MAP leading to better clinical improvement [91]. A study of the blood pressure target in patients with septic AKI with a history of hypertension included 26 patients with a history of chronic hypertension and AKI within the first 24 h of septic shock in two ICUs at the University Hospital of Bordeaux, and the MAP target of 80–85 mmHg was associated with more urine output and lower serum creatinine compared with 65–70 mmHg, confirming the need for higher blood pressure in this group of patients to improve renal perfusion [92]. In a multicenter, open-label trial in France, we randomly assigned 776 patients with septic shock to resuscitation with a mean arterial pressure target of 80 to 85 mmHg (high target group) or 65 to 70 mmHg (low target group). Patients in the high-target group required less renal replacement therapy than those in the low-target group, although the difference in mortality was not associated [93].

In addition to MAP, mean perfusion pressure (MPP) is often used as a clinical indicator of blood pressure. It refers to the difference between the MAP and central venous pressure (CVP) [94]. In a single-center retrospective study conducted in the ICU of Guy’s and St. Thomas’ NHS hospitals, of 2118 ICU patients, 790 patients (37%) developed AKI stage I. AKI I patients with MPP ≤ 59 mmHg were at a significantly increased risk of progression to AKI stage III, and the study concluded that MPP ≤ 59 mmHg was independently associated with AKI progression [95]. In another retrospective analysis of 107 patients hospitalized for infectious shock between August 2010 and June 2013 in the ICU of the tertiary referral university hospital in Melbourne, 55 (51.4%) of whom developed severe AKI, the median MPP decline ratio from premorbidities was 29% in patients who developed severe AKI compared with 24% in non-AKI patients, indicating that a greater MPP decline ratio is associated with the development of septic AKI [96]. A prospective, open-label, before-and-after controlled study conducted in two tertiary ICUs in Australia compared the current standard care and the individualized blood pressure target strategies by screening 62 shocked ICU patients receiving vasopressor medication separately. There was a lower incidence of new significant AKI in the individualized intervention group, indicating that setting an individualized blood pressure target during ICU boosters may prevent AKI and reduce mortality [97]. Taken together, we suggest that clinicians must personalize the management of the patient and could attempt higher blood pressure values in specific cases. Start by targeting 65 mmHg; consider targeting up to 85 mmHg in patients with a previous history of hypertension and monitoring the microcirculation (skin peripheral perfusion) lactate level and urine output to individualize BP targets using fluid infusion and vasoactive drugs.

**Table 2 jcm-12-01018-t002:** Recommendations for sepsis blood pressure targets by study.

Nation	Time	Type of Study	Number of Cases	Indicator	Target	Reference
U.S.	January 2007 to January 2009	Retrospective study	233	MAP	Prehospital MAP	[90]
France	October 2007 to April 2009	Prospective study	217	MAP	72–82 mmHg	[89]
U.S.	2010 to 2016	Retrospective study	8782	MAP	Much higher than 65 mmHg	[86]
Finland	September 2011 to February 2012	Prospective study	423	MAP	Above 73 mmHg	[87]
China	January 2015 to December 2016	Retrospective study	63	MAP	Above 77 mmHg	[88]
France	August 2016 to July 2017	Prospective study	26 (with a history of hypertension)	MAP	80–85 mmHg	[92]
U.K.	July 2007 to June 2009	Retrospective study	790	MPP	Above 60 mmHg	[95]

MAP, mean arterial pressure; MMP, mean perfusion pressure; SBP, systolic blood pressure.

## 6. Replacement Therapy

Kidney replacement therapy is an effective method to treat internal environment disorders caused by kidney failure in septic patients. It also removes toxins to replace renal function, improves blood pressure by removing inflammatory mediators, and helps ultrafiltrate the overload fluid in oliguric AKI [98,99]. At present, there are many clinical kidney replacement treatment modes, so it is necessary to choose the appropriate treatment mode according to the treatment purpose and condition.

Intermittent hemodialysis (IHD) and continuous renal replacement therapy (CRRT) are often used to address fluid overload, and CRRT was previously thought to be advantageous in septic hemodynamically unstable AKI patients [100,101]. More recently, however, it has been suggested that IHD can also reduce hemodynamic instability and improve prognosis if prescribed accurately [102,103]. In addition, IHD has the advantages of being more practical and cost-effective, eliminating the need for anticoagulation and reducing the risk of bleeding.

Other RRT modalities, such as continuous low-efficiency daily dialysis and prolonged intermittent kidney replacement therapy, have shown both hemodynamic stability and cost-effectiveness and are recommended for use in septicemic patients [104]. Hypotension occurs frequently in patients undergoing CRRT and is independently associated with mortality [105,106]. In a retrospective analysis of 2292 AKI patients undergoing CRRT at three referral hospitals, low MAP at the start of CRRT was associated with high mortality, especially when it was <82.7 mmHg [107]. Therefore, blood pressure management in patients during RRT treatment is even more important, which often requires larger doses of pressor medication because of the clearance of replacement therapy. In terms of alternative therapy doses, a meta-analysis of eight prospective randomized controlled trials found that higher-intensity RRT did not reduce mortality and may actually delay renal recovery compared with standard-intensity RRT [108]. In terms of the timing of treatment, Stéphane Gaudry et al. randomized 620 patients with severe AKI to early RRT treatment or delayed treatment. There was no significant difference in mortality between the two groups, even though delayed treatment avoided the need for kidney replacement therapy in a significant number of patients [109]. However, it has also been found that early initiation of RRT to clear fluid overload is more beneficial to renal function recovery in patients with septic AKI compared with patients with delayed initiation of RRT [110].The Second Affiliated Hospital of Guangzhou Medical University, Guangdong, China, is initiating a large, multicenter, prospective, randomized trial on the timing of initiation of continuous renal replacement therapy for acute kidney injury associated with sepsis in the intensive care unit (ClinicalTrials.gov identifier: NCT03175328); the study will enroll 460 patients with KDIGO 2 sepsis AKI from multicenter China, whose findings will help clinicians choose the right time to initiate CRRT.

Some advanced RRT filters have been shown to be effective in removing proinflammatory cytokines and endotoxins, such as the oXiris blood filter, which effectively removes endotoxins, tumor necrosis factor-α, interleukin-6, interleukin-8, and interferon gamma in patients with sepsis [111]. Two French centers reported experience with the oXiris blood filter in patients with septic shock. Studies found a relative reduction in norepinephrine infusion and an improvement in hemodynamic status, but kidney benefit was inconclusive [112]. High-volume peritoneal dialysis (HVPD) may be considered an alternative form of RRT in AKI [113]. However, it lacks competent control of fluids and can be a backup option for a few special cases.

## 7. Conclusions

Blood pressure control in sepsis directly affects the function of multiple organs, including the kidney, as well as the survival rate of patients. It is one of the difficult problems faced by clinicians. In septic hypotension, abnormal microcirculatory shunts leading to renal medullary ischemia and hypoxia are important pathological processes in AKI, and improving microcirculatory perfusion is the goal-oriented approach to improve renal function. Due to the inconsistency in perfusion pressure, the MAP target for patients with septic AKI is recommended to be set to individualize the blood pressure target based on patient-specific conditions, and more research is needed to determine whether a higher blood pressure is needed. Fluid resuscitation is the primary means of improving hypotensive shock, but decreased urine output after renal impairment often leads to fluid overload and requires RRT to maintain positive volume balance. Poor fluid resuscitation requires the addition of pressor drugs to constrict blood vessels to improve the blood pressure and improve the perfusion to all organs. However, more research evidence is needed to provide a clinical basis for patient treatment, including fluid and vasopressor choice, alternative treatment options, and so on, to achieve better blood pressure management goals, improve kidney function, and reduce mortality.

## Figures and Tables

**Figure 1 jcm-12-01018-f001:**
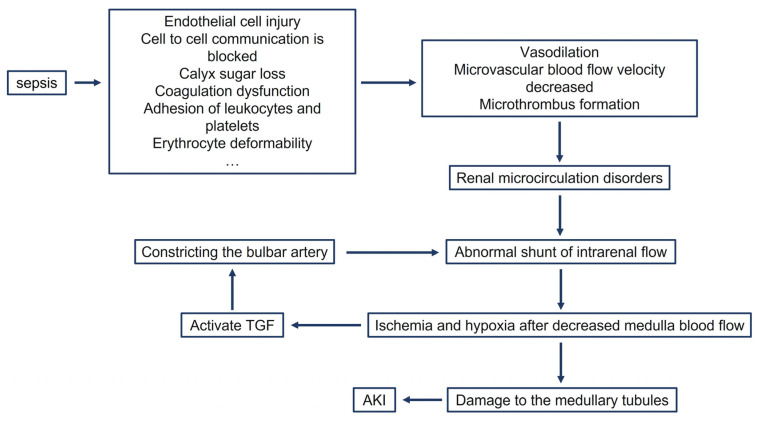
Pathophysiological mechanism of microcirculatory dysfunction affecting renal function in sepsis. TGF, tubuloglomerular feedback mechanism.

## Data Availability

Not applicable.

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
