# Peer review of "Comprehensive Management of Blood Pressure in Patients with Septic AKI"

_jcm, 2023, doi:10.3390/jcm12031018_

Round 1

Reviewer 1 Report

JCM review of Blood pressure management and prognosis of acute kidney injury in sepsis

The authors present a narrative review of the mechanisms and the management of renal vascularization in patients with sepsis and septic shock, with important messages to improve patients’ survival and long-term renal outcomes. The present review of the literature is not exhaustive enough and requires corrections in the interpretation of some articles and in key messages depending on their scientific evidence and medical implications.

Major comments:

-       The title of the article is too restrictive, the review focuses on the management of renal vascularization in septic patients, not only on blood pressure targets.

-       It seems more pragmatic and pedagogic to me to better distinguish the macrovascular and microvascular areas; either when talking about the disorders caused by septic conditions and about the therapeutic tools available in their management. I would rather read in the abstract and then in the order of the chapters: a) mechanisms leading renal injury in sepsis, renal hypoperfusion and microvascular disorders; b) fluid management; vasopressor use; c) blood pressure targets; d) assess and improve microcirculation e) and finally kidney replacement therapy.

-       In microcirculatory disorders: I assume that this would be redundant with the previous comment, but it seems difficult to me, to speak about microvasculature disorders without speaking of microvascular considerations. A word about afferent and efferent arteriola and their sensitivity to angiotensin II is required in this section.

-       In blood pressure target: “began to be significant for AKI at 85 mmHg’’ this is wrong, this is 65mmHg.

-       About quotation 42, what is written is wrong, a 80-85mmHg target MAP was associated with a improved kidney function markers than a 65-70 target. Please be precise on what is written in the manuscript and give the right information when you quote a paper.

-       You should absolutely quote the SEPSISPAM study, this is the biggest trial ever done on the blood pressure targets in septic shock: 10.1056/NEJMoa1312173

-       In fluid management: I totally disagree with this affirmation ‘’ Several studies have shown that albumin improves survival and prognosis and reduces renal impairment in patients with septic shock’’ The only randomized control trial in septic shock patients with albumin versus placebo, funded by the industry and conducted by Jean-Paul Mira was never published, the abstract has been shown in congress and it was negative. Albumin does not prevent AKI in septic patients and never prevented death. Quotation 57 is not a study and 58

-       For this message: “Collectively, it can be argued that diuretics should not be used to treat AKI, except for the treatment of fluid overload’’ please be more affirmative, the 3 articles quoted before are old-fashioned, this is now of medical evidence that diuretics and fluid removal is detrimental at the acute phase of sepsis, and that when patients are stabilized, fluid overload is detrimental and should be removed. Use the articles from Manu Malbrain in Brussels about the 4 phases of shock, resuscitation optimization stabilization and evacuation.

-       In vasopressor: Please refer to the international guidelines from the surviving sepsis campaign and do not quote little articles from neonatal reanimation to say that dopamine is equal to epinephrine… For the norepinephrine that is effectively recognized by the experts as the main molecule for the treatment of shock, because it has a B-adrenergic effect that helps the heart to contract, in contrast to Ang-II or vasopressin that may increase septic cardiomyopathies by increasing the left ventricle afterload without helping the heart to contract. You can not assume ‘’Norepinephrine… is the medication of choice in hypotension’’ and 5 lines later ‘’ the proposed use of norepinephrine in patients with septic AKI is controversial’’. Please read carefully what it written in the surviving sepsis campaign to analyze the literature.

-       In renal replacement therapy: It is right that kidney replacement therapy is able to remove some cytokines in septic patients. But the important point is: it has never been shown that removing this cytokines improves patient’s outcomes. Some studies have even shown that early replacement therapy with high flux may be detrimental. Please just mention that all the efforts made to show that removing cytokines could be beneficial for patient have failed these last 20years (maybe because there are too many cytokines with either pro-inflammatory effects or anti-inflammatory effects). Neither the intensity of dialysis (10.1093/ndt/gfx308) nor earlier KRT (10.1056/NEJMoa1603017) nor other trials with polymixin coated membranes or absorption of cytokines have been conclusive.

-       “it is recommended that RRT should be started early…” This is totally wrong, once again don’t quote little papers, 10.1056/NEJMoa1603017 - 10.1016/S0140-6736(21)00350-0.- 10.1056/NEJMoa2000741). For 2 years, the commonly accepted criteria to start dialysis in ICU patients (not talking about intoxications) are: AKI stage III of the KDIGO classification AND medication resistant hyperkaliemia (>6mmol/L) OR medication resistant pulmonary oedema OR medication resistant metabolic acidosis (pH<7.15), OR 72h oligo-anuria OR urea level > 40mmol/L. All is written here: 10.1056/NEJMra2104090.

-       CRRT is not better than intermittent dialysis when it is correctly prescribed (10.1164/ajrccm.162.1.9907098 and once again 10.1056/NEJMra2104090). These papers are stronger than the quotations from 102-106.

-       111 and 112, no evidence for patient benefits in randomized control studies (except some in Japan but never in other part of the world). Once again please just mention that all the efforts made to show that removing cytokines could be beneficial for patient have failed these last 20years.

-       In the conclusion: ‘’ the MAP target for patients with septic AKI are recommended to be higher than those for patients with simple sepsis’’ Which recommendation?

-       ‘’ However, more research evidence is needed to provide physicians with a clinical basis for the choice of drugs and treatment modalities to achieve better blood pressure management goals, improve renal function, and reduce mortality.’’ It seems that nothing has been made in the field to guide clinicians, please give tips to cure the patients.

Minor comments:

-       In the abstract ‘’sepsis’’ (sepsis) is useless.

-       In the abstract ‘’Furthermore, it has been found that some patients with sepsis-associated AKI have a previous history of hypertension, and a small change in blood pressure will have an adverse effect on renal function.’’ This sentence sounds strange of course patients with previous hypertension have sepsis and possibly AKI. « In patients with a previous history of hypertension, small changes in blood pressure may be even more deleterious for kidney function » would be more appropriate.

-       The first sentence of the introduction, at my point of view, a dysregulated host response leads to septic shock, not to sepsis.

-       In the introduction: ‘’but no tubular necrosis’’ this is wrong, from Lerolle et al. « Proximal and distal tubules in all patients showed the changes usually associated with acute tubular injury: loss of brush border, frank necrosis and dilatation of the tubules with variable flattening of the cytoplasm.’’ These are acute tubular necrosis lesions.

-       In the introduction: ‘’This is associated with systemic microcirculatory … patients with AKI or at risk of AKI’’ This sentence is far too long, and the beginning of the sentence is redundant.

-       Please use kidney instead of renal as possible (example: kidney failure, kidney replacement therapy)

-       In figure 1: ‘’…’’ seems not appropriate, you should give exhaustive information in this review.

-       In blood pressure target: at the 13th line is written “garget” instead of target.

-       In the title of the chapter replAcement therapy, the “a” is missing.

-       The study from the mayo clinic “lowest incidence of AKI’’ lowest and of AKI have wrong police.

-       Against the quotation 40, the results of this paper: 10.1186/s13613-021-00969-4; please discuss these conflicting results.

-       For quotation 44, give consistency with the MPP target, <60 or =/<59mmHg.

-       At the end of blood pressure target, you mention that clinician should individualize blood pressure target, but tips for individualization should be mentioned. Start by targeting 65mmHg; consider targeting up to 85mmHg in patients with a previous history of hypertension and monitoring microcirculation (skin peripheral perfusion) lactate level and urine output to individualize BP targets using fluid infusion and vasoactive drugs.

-       I would not mention pediatric ICU studies, they seem inappropriate in this review. Or build a special paragraph with more documentation and underlying the pediatric specificities in the field.

-       In fluid management: Quotation 49 has no relation with the sentence. Quote instead the recommendation from the surviving sepsis campaign.

-       Redundance around quotation 59.

-       In Renal replacement therapy: ‘’ Renal replacement therapy is an effective treatment for septic AKI’’ it is an effective treatment to avoid dying by hyperkaliemia, but it does not treat septic AKI.

Author Response

Thank you very much for the reviewer's comments. My reply is as follows:

Major comments:

  1. The title of this paper is blood pressure Management. In addition to blood pressure targets, this paper summarizes the management of fluids, drugs and renal replacement therapy, and does not exceed the title limit.
  2. The focus of this review is to provide management objectives and methods for blood pressure management in patients with septic AKI, so the idea is to pay attention to the causes of blood pressure (microcirculation mechanism), blood pressure target values, and fluid, drug and alternative therapy.
  3. I have included the afferent and efferent arterioles and their sensitivity to angiotensin II in the drug section.

4-13、15:I have corrected them, and has been marked red in the text.

14: “The MAP target for patients with septic AKI are recommended to be higher than those for patients with simple sepsis” This recommendation comes from PMID: 34553274 and 24635770.

Minor comments:

1-10、12-17: I have corrected them, and has been marked red in the text.

11: Although higher CVP and lower MPP were associated with AKI progression after cardiac surgery, relative hypotension was not associated with AKI progression, 10.1186/s13613-021-00969-4 said. However, these findings are based on exploratory surveys, and further research is needed to confirm them. the quotation 40 considers it necessary to target pre-admission MAP to prevent the occurrence of sepsis AKI. First of all, the two causes are different, respectively, sepsis and heart surgery. Both agree that lower MPP or MAP is associated with AKI. So the results are not contradictory.

Reviewer 2 Report

In the current review by Deng et al., authors have summarized blood pressure management and effect of microcirculation in sepsis induced AKI. The authors have covered aspects of various patho physiological changes affecting the microvasculature which can lead towards AKI during sepsis. Authors have addressed other factors like blood pressure, vasopressors, fluid management and other clinical findings associated with clinical management with appropriate literature. Each topic is well covered and cited with relevant literature. However, I have certain queries/suggestions:

1- Talking about microcirculation and AKI, one very important aspect of lymphatic flow/vasculature has not been addressed anywhere in the review. Recent reports are suggesting an important part of renal pressure, renal flow and immune cells are regulated by lymphatic vessels which also plays an important role in blood pressure management by immune cell infiltration. I am just thinking if authors have any insights in this direction.

  • PMID: 22179226, 34806312, 31913221

2- Major aspects of the current review has been described/published before as well (specially by Evans group- PMID: 29908046, PMID: 31836037PMID: 31566903,  could authors justify novelty in the current review. 

Author Response

Thank you very much for the reviewer's comments. My reply is as follows:

1.This review is based on the clinical blood pressure management of patients with septic AKI, and mainly focuses on the influence of patients' major arterial blood pressure on renal function. However, renal lymphatic vessels are not associated with major arterial blood pressure, so this aspect is not discussed in this paper. Studies have mentioned that renal immune cell accumulation is associated with hypertension, but this review focuses on the management of septic hypotensive shock. I will summarize and discuss this aspect if I write about the renal mechanisms associated with hypertension in the future.

2.

These three reviews only discussed the influence of renal microcirculation hypoxia on renal function and antihypertensive drugs. This review summarized and discussed the pathological mechanism of improving renal function by blood pressure management, clinical target value, body fluid management, drug selection and renal replacement therapy, which systematically improved the reference basis of great value for clinical blood pressure management. What's new is that it's more clinical, more comprehensive.

Round 2

Author Response

-The title of the article is too restrictive, the review focuses on the management of renal vascularization in septic patients, not only on blood pressure targets.

I have changed the title to " Comprehensive management of blood pressure in patients with septic AKI"

-It seems more pragmatic and pedagogic to me to better distinguish the macrovascular and microvascular areas; either when talking about the disorders caused by septic conditions and about the therapeutic tools available in their management. I would rather read in the abstract and then in the order of the chapters: a) mechanisms leading renal injury in sepsis, renal hypoperfusion and microvascular disorders; b) fluid management; vasopressor use; c) blood pressure targets; d) assess and improve microcirculation e) and finally kidney replacement therapy.

I have reorganized the chapters according to the review comments.

-You should absolutely quote the SEPSISPAM study, this is the biggest trial ever done on the blood pressure targets in septic shock: 10.1056/NEJMoa1312173

I have corrected them, and has been marked red in the text.

-For this message: “Collectively, it can be argued that diuretics should not be used to treat AKI, except for the treatment of fluid overload’’ please be more affirmative, the 3 articles quoted before are old-fashioned, this is now of medical evidence that diuretics and fluid removal is detrimental at the acute phase of sepsis, and that when patients are stabilized, fluid overload is detrimental and should be removed. Use the articles from Manu Malbrain in Brussels about the 4 phases of shock, resuscitation optimization stabilization and evacuation.

I have corrected them, and has been marked red in the text.

-In vasopressor: Please refer to the international guidelines from the surviving sepsis campaign and do not quote little articles from neonatal reanimation to say that dopamine is equal to epinephrine… For the norepinephrine that is effectively recognized by the experts as the main molecule for the treatment of shock, because it has a B-adrenergic effect that helps the heart to contract, in contrast to Ang-II or vasopressin that may increase septic cardiomyopathies by increasing the left ventricle afterload without helping the heart to contract. You can not assume ‘’Norepinephrine… is the medication of choice in hypotension’’ and 5 lines later ‘’ the proposed use of norepinephrine in patients with septic AKI is controversial’’. Please read carefully what it written in the surviving sepsis campaign to analyze the literature.

I have corrected them, and has been marked red in the text.

-In renal replacement therapy: It is right that kidney replacement therapy is able to remove some cytokines in septic patients. But the important point is: it has never been shown that removing this cytokines improves patient’s outcomes. Some studies have even shown that early replacement therapy with high flux may be detrimental. Please just mention that all the efforts made to show that removing cytokines could be beneficial for patient have failed these last 20years (maybe because there are too many cytokines with either pro-inflammatory effects or anti-inflammatory effects). Neither the intensity of dialysis (10.1093/ndt/gfx308) nor earlier KRT (10.1056/NEJMoa1603017) nor other trials with polymixin coated membranes or absorption of cytokines have been conclusive.

I have corrected them, and has been marked red in the text.

-In the conclusion: ‘’ the MAP target for patients with septic AKI are recommended to be higher than those for patients with simple sepsis’’ Which recommendation?

I have corrected them, and has been marked red in the text.

-In the introduction: ‘’but no tubular necrosis’’ this is wrong, from Lerolle et al. « Proximal and distal tubules in all patients showed the changes usually associated with acute tubular injury: loss of brush border, frank necrosis and dilatation of the tubules with variable flattening of the cytoplasm.’’ These are acute tubular necrosis lesions.

My expression here is misleading, is to express the damage is not uniform, the article has been changed. I have corrected them, and has been marked red in the text.

Reviewer 2 Report

The authors have addressed my queries and suggestions promptly and I have no further comments to make.

Author Response

Thanks!